# Are There Differences in the Reaction of the Light-Tolerant Subgenus *Pinus* spp. Biomass to Climate Change as Compared to Light-Intolerant Genus *Picea* spp.?

**DOI:** 10.3390/plants9101255

**Published:** 2020-09-23

**Authors:** Vladimir A. Usoltsev, Hui Lin, Seyed Omid Reza Shobairi, Ivan S. Tsepordey, Zilin Ye

**Affiliations:** 1Research Center of Forestry Remote Sensing & Information Engineering, Central South University of Forestry and Technology, Changsha 410004, China; usoltsev50@mail.ru (V.A.U.); Omidshobeyri214@gmail.com (S.O.R.S.); loyzer@163.com (Z.Y.); 2Faculty of Forestry, Ural State Forest Engineering University, Sibirskiy Trakt, 37, 620100 Yekaterinburg, Russia; 3Botanical Garden of Ural Branch of RAS, Department of Forest Productivity, ul. 8 Marta, 202a, 620144 Yekaterinburg, Russia; common@botgard.uran.ru; 4Key Laboratory of Forestry Remote Sensing Based Big Data & Ecological Security for Hunan Province, Changsha 410004, China; 5Key Laboratory of State Forestry Administration on Forest Resources Management and Monitoring in Southern Area, Changsha 410004, China; 6Changsha Changchang Forestry Technology Consulting Co., Ltd., Changsha 410004, China

**Keywords:** hydrothermal gradients, stand biomass, regression models, biomass equations, January temperature, annual rainfall

## Abstract

Currently, the problem of the impact of climate change on the productivity of forest ecosystems and their carbon-depositing capacity is far from being solved. Therefore, this paper presents the models for the stand biomass of the two-needled subgenus’ (*Pinus* spp.) and the genus *Picea* spp.’s trends along the trans-Eurasian hydrothermal gradients, designed for pure stands in a number of 2110- and 870-sample plots with *Pinus* and *Picea* correspondingly. It was found that in the case of an increase in mean winter temperatures by 1 °C, pine and spruce respond by increasing the biomass of most components, and in the case of an increase in the annual sum of precipitation by 100 mm, the total, aboveground, stem and root biomasses of pine and spruce react the same way, but crown biomass reacts in the opposite way. Therefore, all identified trends are species-specific.

## 1. Introduction

In recent decades, the problem of the impact of climate change on the productivity of forest ecosystems and their carbon-depositing capacity has become of great importance [1]. However, there are many uncertainties in this problem [2,3]. There is evidence that changes in rainfall ability have a stronger short-term effect on forest productivity than changes in warm ability [4]. One of the most significant examples of such evidence is the contradiction of two possible scenarios: on the one hand, an increase in biological productivity due to the potential enrichment of the atmosphere with CO_2_ and its reclamation effect on vegetation, and on the other hand, its decrease due to the loss of stability with a sharp reduction in the adaptation time lag [5,6,7].

In the history of civilization and in its future, forests play an important role, being under the influence of climate fluctuations [8,9,10,11]. The analysis of climate changes by annual tree rings was first applied in the 19th century by A. N. Beketov [12] and F. N. Shvedov [13], and then by V.F. Klyuchnikov [14], A. Douglas [15], S.G. Zaozersky [16] and A.P. Tolsky [17]. Later, this method was developed in a new scientific direction, namely, dendro-climate-chronology [18,19], now successfully used in the analysis of past climate changes and its forecasts for the future. It is known that the variability of the width of the annual ring, which records the tree’s response to the environment, is largely determined by the cycles of solar activity and other exogenous and endogenous factors [15,20]. However, in conditions of excessive moisture, where the limiting factor is the lack of oxygen in the soil, the cyclical growth is largely explained by the hydrological regime of the rhizosphere [21]. At the other extreme, namely the lack of moisture in the steppe conditions, the tree’s sensitivity to precipitation increases, expressed in increased variability in the width of the annual rings [22]. Recently, research on dendrochronology and climatochronology has reached a global level using the International Tree-Ring Data Bank [23].

The dependence of the radial growth of Norway spruce, Scots pine and European beech trees on the climate and water balance of the soil was studied using data from 24 sample plots in Germany for the period from 1951 to 2006 [24]. Using a standard multi-factor regression analysis procedure for each tree species, one analyzed the relationship of growth with 30 independent variables that characterize precipitation, air temperature and soil water balance for different months, which explained from 50% to 57% of the total variability of growth. By combining the obtained models with climate forecast data published by the IPCC expert group for the period up to 2100, a forecast of the dynamics of the growth of three tree species up to 2100 was made [24]. It turned out that for spruce, environmental conditions become more and more unfavorable over time, which leads to a gradual decrease in growth. For Scots pine and (with rare exceptions) for beech, the negative effect on the radial growth of simulated climate scenarios and soil water balance could not be detected until 2100 [24]. However, according to more recent research, all European species may face a significant decrease in suitable habitat area [25].

This problem concerns not only radial growth, but also forest biomass production [26,27]. Current research literature reports a somewhat tentative explanation of the impact of warm and rainfall ability on forest biomass in China [28,29], Europe [30] and the United States [31]. The uncertainty of the results is largely due to the narrow range of independent variables used, including climate variables [32].

In relation to a forest ecosystem, the regression model is the result of statistical estimation of the parameters of a system of mathematical expressions that characterize a certain biological concept of the relationship of phenomena. The variability of forest ecosystem parameters can be explained using not one, but several variables that are partially or completely interdependent. Separate modeling of such dependencies leads to the fact that the obtained estimates will not be balanced, or harmonized. Mathematical dependencies combined into a single logically consistent concept form a system of related (compatible, recursive) equations. Their main advantage is the internal consistency of the described chain of equations [33,34,35,36]. Another method of harmonizing the patterns of forest biomass suggests the regionalization of a model by introducing dummy variables [37] encoding the ecoregion or tree species affiliations of the harvest data. Biomass models that include mass-forming indices and dummy variables are known as mixed-effect models [38,39,40,41,42]. However, they have a disadvantage, namely that they take into account regional shifts in the calculated values only by the intercept term, while the regression coefficients are assumed to be unchanged for regions, what is not true.

The two-needled subgenus (*Pinus* spp.) includes about 100 species spread in boreal and mid-latitude zones, and also in the mountain regions of the subtropical zone of the Northern Hemisphere [43]. There are about 10 species in Russia. Of the two-needled pines, the Scots pine (*Pinus sylvestris* L.) is the most common in Eurasia, and among coniferous only larches occupy a bigger area than pines. This is a large evergreen whorl-branching light-demanding tree with a transparent crown. Its needle foliage is adapted to conservative water consumption, tolerates temperatures of −50 °C to +50 °C, and lives for 2–6 years [44,45].

The genus *Picea* spp. consists of many species that are subject to hybridization. The hypothesis of the mountain origin of spruces suggests their origin in the tertiary period in mountainous conditions. An indirect confirmation of this hypothesis is the presence of a surface root system and the timing of seedlings to rotting detritus [46].

The influence of warm and rainfall ability on the biomass change of forest species has been studied in relation to two-needled pine—subgenus *Pinus* spp. [47]. In the mentioned work, though, only transcontinental trends were given, according to which the biomass of stands changes due to both average winter temperatures and average annual precipitation, i.e., it assessed the pine biomass relationship against temperature and precipitation. In addition, the input data included about 30% of the sample plots with the admixture of other species.

The purpose of this study was to show how much the biomass component (stems, roots, foliage, branches) composition of the light-tolerant subgenus *Pinus* sp.’s biomass can change, taking into account a possible increase in temperature by 1 °C at constant precipitation, and with a possible increase in precipitation by 100 mm per year at a constant temperature, and to compare the results with similar data for light-intolerant genus *Picea* spp.

## 2. Material and Methods

Since the response to climate change differs between mixed and pure stands [48], in our comparative study we used harvest data only from pure stands that were selected from our data set [49]; 2110 and 870 samples with *Pinus* and *Picea* biomass data. Unfortunately, the biomass of the roots was collected only from 920 and 515 plots, for *Pinus* and *Picea* respectively. Mostly pure stands with an admixture of other species up to 10% in pine and spruce forests were selected from the database.

In most cases, sample trees were taken on each of the sample plots in numbers from 5 to 10 copies. Then, samples were taken from each biomass component to determine the dry matter content (and from the wood and bark of stems as well, to determine the basic density) after drying the samples at temperatures of 80–100 °C. The quantity of each biomass component per 1 ha was determined by a regression method. Nevertheless, some sampling procedures for estimating the biomass of tree components differed between the studies, since they were performed by representatives of different scientific fields in forestry.

The distribution of data collected from different sites is shown in Figure 1. Each sample area is mapped to the available coordinates for January temperatures and annual rainfall (World Weather Maps [50]).

It was found earlier that when estimating stem biomass growth by using the annual ring width, the greatest contribution to explaining its variability was made by summer temperature, accounting for from 16% of the total dispersion [51] to 50% of the residual one [52]. Moreover, the specificity of the relationship (positive or negative) of stand biomass depends on what intra-annual temperature was taken as a predictor: it was established by Khan et al. [53] that this relationship is positive with the maximum intra-annual temperature, and negative with the minimum and average annual temperature. With an inter-annual time step, the predominant influence of summer temperature is quite normal [54,55]. However, against the background of long-term climatic shifts for decades, the prevailing influence is acquired by winter temperatures [56], bearing in mind that winter temperatures in the Northern Hemisphere increased faster than summer ones during the 20th century [57,58,59,60]. In terms of regression analysis, a weak temporal trend of summer temperatures compared to a steep total variance explained by this regression. Obviously, taking the mean winter temperature as one of the independent variables, we get a more reliable dependence with a higher predictive ability.

The input data matrix was used as a source of data in the common regression analysis, and recursive equations for the different components of the stand biomass were derived [61]:
ln*N* = f {ln*A,* ln(*Tm* + 40), ln*PRm*, [ln(*Tm* + 40)]·(ln*PRm*)} →→ln*V* = f [ln*A,* ln*N*, (ln*A*)(ln*N*), ln(*Tm* + 40), ln*PRm*, [ln(*Tm* + 40)]·(ln*PRm*)} →→ln*Pi* = f {ln*A*, ln*V*, ln*N*, (ln*A*)(ln*N*), ln(*Tm* + 40), ln*PRm*, [ln(*Tm* + 40)]·(ln*PRm*)}(1)


In Equation (1), the designations here and further are as follows: *Pi* is the biomass of the *i*-th fraction, and *Pt*, *Pa*, *Pr*, *Ps*, *Pf* and *Pb* are the biomass of total (above- and underground) wood storey, aboveground storey, underground storey (roots), stems (wood and bark), foliage and branches correspondingly, t per ha; *A* is forest age, years; *V* is volume stock, m^3^ per ha; *N* is tree number, 1000/ha; *PRm* is averaged annual rainfall, _MM_; *Tm* is average temperature in January, °C. The averaged January temperature in the north of Eurasia has the sign “minus”, and it is supplemented by a value 40 (*Tm* + 40). The regression analysis was carried out in Statgraphics software (http://www.statgraphics.com/) with the use of a logarithm transformation corrected by Baskerville [62]. The results of Equation (1) for the different components of the stand biomass are presented in Table 1. The recursive system of Equation (1) was tabulated according independent variables.

## 3. Results and Discussions

From the results of Equation (1)’s tabulation, the values of the component composition of biomass for the age of maturity (100 years) are taken, and 3D graphs showing statistically significant transcontinental trends of biomass are designed for each biomass component in temperature and precipitation gradients (Figure 2 and Figure 3).

We can see significant differences in the changes in the structure of pine and spruce biomass in the precipitation and winter temperature gradients. Pine and spruce show a monotonous increase in all the biomass components in the gradient of temperature increase (Figure 2 and Figure 3), with the exception of the mass of needles in pine stands, which in areas of insufficient moisture (*PRm* = 300 mm) is not increased, but reduced with the transition from cold to warm zones, and the mass of branches remains unchanged. Pine shows a similar pattern only for total, aboveground, stem and underground biomass, and only in areas of sufficient moisture (*PRm* = 900 mm) (Figure 2). In areas of insufficient moisture (*PRm* = 300 mm), the biomass of foliage decreases with the transition from cold to warm zones, and the mass of branches remains unchanged. In the gradient of increasing precipitation, spruce does not show as clear a pattern as in the gradient of increasing temperature. In cold regions (*Tm* = −30 °C), most of the biomass components do not react to changes in precipitation in any way, while in warm regions (*Tm* = 10 °C), as precipitation increases, there is a slight increase in total, aboveground and stem biomass, and, on the contrary, a slight decrease in the biomass of needles and branches (Figure 3). Changes in the pine biomass components in precipitation gradients are more obvious compared to spruce: the total, aboveground, underground and stem biomass decrease in all warm zones, and the mass of foliage and branches decreases only in cold zones (*Tm* = −30 °C), while in zones of high warming (*Tm* = 10 °C) it increases (Figure 2).

In recent decades, scientists want to know how much the structure of forest biomass will change with an air temperature shift, for example, by 1 °C, and with a deviation of precipitation from the usual norm, for example, by 100 mm per year. We cannot directly measure future global situations, but the constructed model gives the answer to such question in relation to forest stands. To do this, we took the first derivative of three-dimensional surfaces (Figure 2 and Figure 3), and got the answer in the form of three-dimensional surfaces divided into plus and minus areas, which correspond to the increase or decrease in the biomass of stands having the age of 100 years for *Pinus* (Figure 4 and Figure 5) and for *Picea* (Figure 6 and Figure 7).

Figure 4 and Figure 6 show the change in *Pinus* and *Picea* biomass (Δ, %) with a temperature increase of 1 °C in areas characterized by different temperature and rainfall ratios. They demonstrate a common regularity in *Pinus* and *Picea* on the Eurasian scale: the increase (the location of the increment percentage surface above the zero plane) in different biomass components with the temperature increase of 1 °C in all the temperature zones, and with different precipitation levels (the location of the increment percentage surface of all the components is above the zero plane). As we move from cold (*Tm* = −30 °C) to warm areas (*Tm* = 10 °C), the percentage of this increase decreases (Figure 4 and Figure 6). The exception is the biomass of needles, which in areas of insufficient moisture under the same conditions does not increase, but decreases in all temperature zones.

The biomass of pine and spruce (Δ, %), however, reacts differently to changes in precipitation. If the total, aboveground, stem and root biomass in pine decrease (the location of the increment surface is below the zero plane) as a result of the expected increase in annual rainfall by 100 mm, regardless of regional precipitation and temperature, then spruce shows this trends only in cold temperature zones (*Tm* = −30 °C), while in warm (*Tm* = 10 °C) it increases (the location of the increment surface is above the zero plane; the exception is the biomass of roots in spruce). Opposite trends in pine and spruce under the same conditions are observed for the biomass of both needles and branches: in cold regions, their biomass decreases in pine and increases in spruce, and in warm regions, the contrary is true (Figure 5 and Figure 7).

The proportions of the contributions of independent variables to explaining the variability of the dependent variables in Equation (1) are shown in Table 2. It shows the contribution of various independent variables to the explanation of the variability of the desired indices of biomass. We can see that mass-forming variables explain on average about 87% and 93% of the variability of all the biomass components in *Pinus* and *Picea*, correspondingly, including 73% and 80% of the contribution from the stem volume. Climate variables explain only about 13% and 7% of the total biomass variability in *Pinus* and *Picea*, correspondingly, i.e., 7 and 13 times less than the mass-forming variables in *Pinus* and *Picea* correspondingly. The least sensitive to climate change is the biomass of stems (0.9–2.9%).

The patterns of biomass amount change under assumed changed climatic conditions (Figure 2, Figure 3, Figure 4, Figure 5, Figure 6 and Figure 7) seem to be hypothetical. They reflect the long-term adaptive responses of forest ecosystems to regional climate, and ignore the rapid trends of current climate changes, which place serious constraints on forests adaptations to new climatic conditions [51,63,64,65,66,67,68]. The law of limiting factors [69,70] works well in stationary conditions. With the rapid change in limiting factors (such as air temperature or precipitation), forest ecosystems are in a transitional (non-stationary) state, in which some factors that are still not significant may come to the fore, and the end result may be determined by other limiting factors [71,72]. Increasing tree diversity is recommended as the best option for an uncertain future [73]; otherwise we may also lose valuable genetic pools [74].

As can be seen from Table 2, the contribution of climate variables in comparison to the variables of the ages and morphological structures of stands is quite small, and this gives reason to emphasize once again the preliminary nature of the expected shifts in the component structure of the biomass of the studied stands. It is obvious that as the harvest database is filled and the existing “white spots” in their availability for certain geographical areas (ecoregions) are closed, the level of uncertainty will decrease. In this regard, it should be noted that the results of modern process-based models are also quite contradictory, and show no less uncertainty in predicting changes in vegetation cover under the influence of climate [75,76]. One possible reason for uncertainty is that process-based models based on tree biological peculiarities are actually correlative, since the responses of the forest ecosystem to environmental changes include not only hydrothermal conditions, but also genetically regulated changes that are difficult to account for [77].

The main pool of our harvest data on forest biomass in Eurasia was obtained from the 1970s to the 1990s, and the climate maps used cover the period of the late 1990s to the early 2000s. Some discrepancy between the two time periods may cause possible biases in the results obtained, but for such a small time difference in the initial data, the inclusion of compensatory mechanisms or phenological shifts in forest communities is unlikely [67,78]. There is an uncertainty in assessing the impact of phenology on the biological productivity of stands, established for the cherry oak in the south of Russia: while the assessment of the biomass of oak stands did not reveal differences between the phenologic varieties of oak, the assessment of net primary production shows a 1.6-fold advantage of the late-blooming variety over the early-blooming one [79].

## 4. Conclusions

This paper presents the models for the stand biomass of the two-needled subgenus’ (*Pinus* spp.) and the genus *Picea* spp.’s trends along the trans-Eurasian hydrothermal gradients, designed for pure stands in numbers of 2110- and 870-sample plots with *Pinus* and *Picea* correspondingly. It was found that in the case of an increase in mean winter temperatures by 1 °C, pine and spruce respond by increasing the biomass of most components, and in the case of an increase in the annual sum of precipitation by 100 mm, the total, aboveground, stem and root biomass of pine and spruce react the same way, but crown biomass reacts in the opposite way.

Taking into account the stated methodological and conceptual uncertainties, the results presented in this study should be considered as preliminary ones, and our outputs represent just an example of model sensitivity to changing climatic conditions. The development of such models for the other species of Eurasia will allow us to predict changes in the productivity of the forest ecosystems of Eurasia under assumed climate change.

## Figures and Tables

**Figure 1 plants-09-01255-f001:**
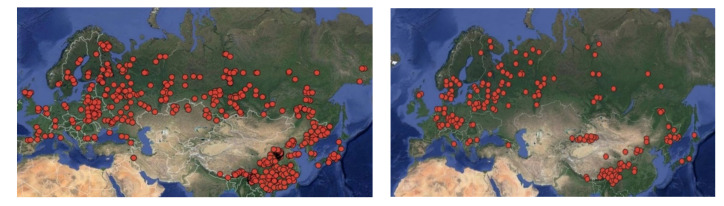
The distribution of 2110 sampling sites of *Pinus* (on the **left**) and 870 sampling sites of *Picea* (on the **right**) on the territory of Eurasia.

**Figure 2 plants-09-01255-f002:**
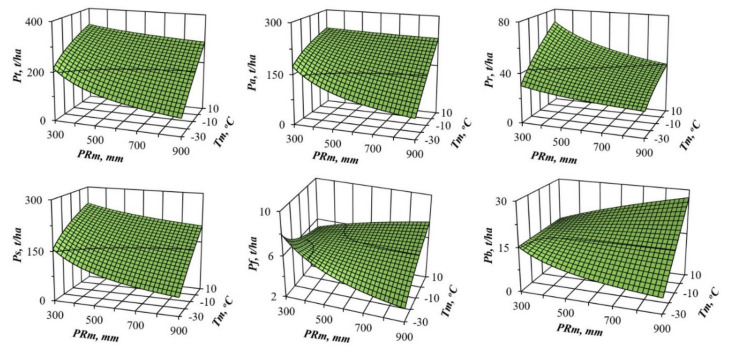
The relation of pine biomass to the averaged temperature of January (*Tm*) and averaged precipitation (*PRm*) at the stand age of 100 years. Designations: *Pt*, *Pa*, *Pr*, *Ps*, *Pf* and *Pb* are the biomass of total (above- and underground) wood storey, aboveground storey, underground storey (roots), stems (wood and bark), foliage and branches correspondingly, t per ha.

**Figure 3 plants-09-01255-f003:**
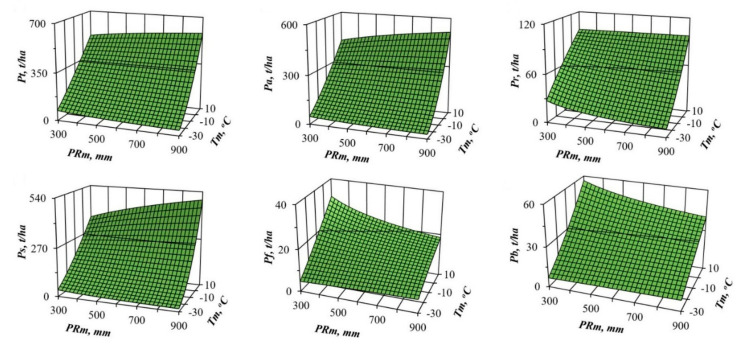
The relation of spruce biomass to the averaged temperature of January (*Tm*) and precipitation (*PRm*) at the stand age of 100 years. Designations: *Pt*, *Pa*, *Pr*, *Ps*, *Pf* and *Pb* are the biomass of total (above- and underground) wood storey, aboveground storey, underground storey (roots), stems (wood and bark), foliage and branches correspondingly, t per ha.

**Figure 4 plants-09-01255-f004:**
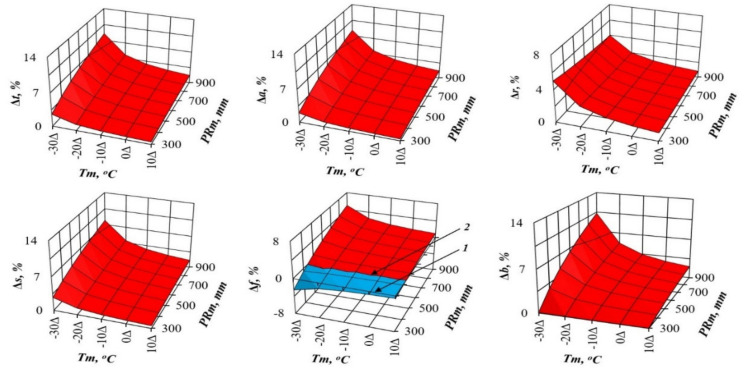
The theoretical changes in pine biomass in relation to the assumed warming increase of 1 °C based on the model (1) for the stands aged 100 years. Here and further: 1—the plane corresponding to zero change in biomass at the expected temperature increase by 1 °C; 2—the line of the differing of positive and negative changes in biomass (Δ, %) with an expected increase in temperature of 1 °C.

**Figure 5 plants-09-01255-f005:**
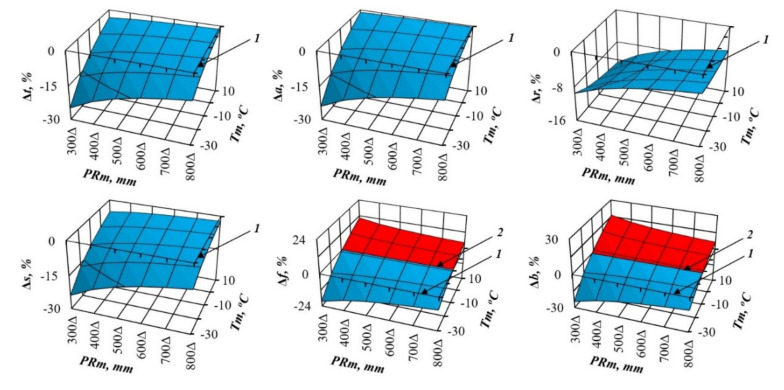
The theoretical changes in pine biomass in relation to the assumed rainfall of 100 mm based on the model (1) for the stands aged 100 years.

**Figure 6 plants-09-01255-f006:**
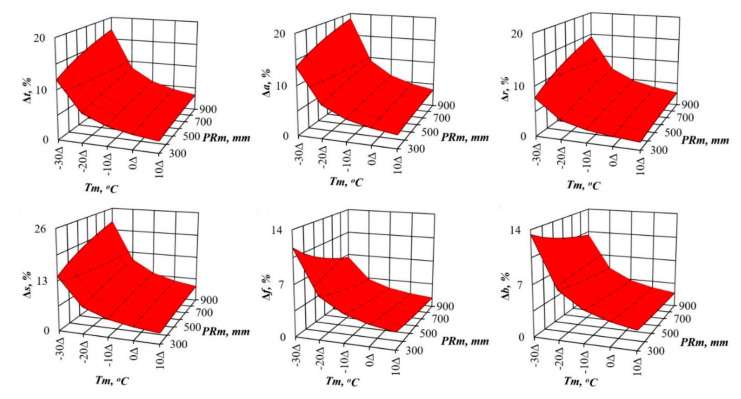
The theoretical changes in spruce biomass in relation to the assumed warming increase of 1 °C based on the model (1) for the stands aged 100 years. Designations: Δ*t*, Δ*a*, Δ*r*, Δ*s*, Δ*f* and Δ*b* are percentages of the biomass of total (above- and underground) wood storey, aboveground storey, underground storey (roots), stems (wood and bark), foliage and branches correspondingly, with a temperature increase of 1 °C.

**Figure 7 plants-09-01255-f007:**
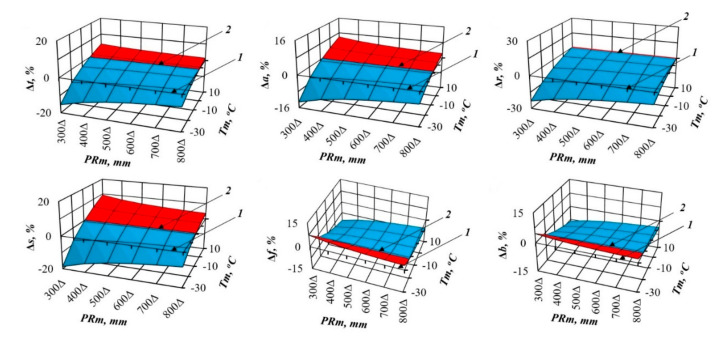
The theoretical changes in spruce biomass in relation to the assumed rainfall of 100 mm based on the model (1) for the stands aged 100 years. Designations: Δ*t*, Δ*a*, Δ*r*, Δ*s*, Δ*f* and Δ*b* are percentages of the biomass of total (above- and underground) wood storey, aboveground storey, underground storey (roots), stems (wood and bark), foliage and branches correspondingly, with a rainfall of 100 mm.

**Table 1 plants-09-01255-t001:** Characteristic of Equation (1) for forest stands of the subgenus *Pinus* spp., and the genus *Picea* spp. in Eurasia.

ln(*Y*) ^(1)^	a_0_ ^(2)^	ln*A*	ln*V*	ln*N*	(ln*A*)·(ln*N)*	ln(*Tm* + 40)	ln*PRm*	[ln(*Tm* + 40)]·(ln*PRm*)	adjR^2 (3)^	SE ^(4)^
*Pinus*										
ln(*N*)	2.8168−	−1.0696	-	-	-	1.9165	0.5011	-0.3577	0.566	0.72
ln(*V*)	16.4304	0.7200	-	−0.7996	0.2065	−3.3579	−2.5007	0.6225	0.472	0.69
ln(*Pt*)	1.8338	0.0782	0.8291	0.0367	−0.0031	−0.5645	−0.3541	0.1157	0.944	0.16
ln(*Pa*)	3.4809	0.0441	0.8221	0.0924	−0.0240	−1.1030	−0.5944	0.1931	0.952	0.17
ln(*Pr*)	−12.6886	0.1991	0.7013	−0.1314	0.0597	3.2874	1.7271	−0.5006	0.770	0.33
ln(*Ps*)	0.2311	0.1283	0.9346	−0.0173	0.0137	−0.4483	−0.2603	0.0901	0.966	0.17
ln(*Pf*)	11.8492	−0.3495	0.4313	0.1311	−0.0289	−3.6140	−1.6841	0.5555	0.424	0.36
ln(*Pb*)	9.6761	−0.1607	0.5734	−0.0327	−0.0154	−3.3278	−1.6096	0.5548	0.650	0.39
*Picea*										
ln(*N*)	−11.5190	−1.0091	-	-	-	5.2191	3.0637	−0.9733	0.569	0.61
ln(*V*)	−11.2252	0.8455	-	−1.1457	0.2699	−2.9798	−2.4759	0.7765	0.642	0.59
ln(*Pt*)	0.0180	−0.0646	0.8128	−0.0875	0.0208	0.6708	0.1668	−0.1113	0.970	0.19
ln(*Pa*)	−1.2565	−0.0524	0.8145	−0.0796	0.0224	0.8107	0.2335	−0.1077	0.972	0.17
ln(*Pr*)	4.1101	−0.0090	0.7970	−0.0973	0.0296	−0.7176	−0.6121	0.0666	0.884	0.39
ln(*Ps*)	0.3868	0.0371	0.9481	−0.1493	0.0393	−0.0652	−0.2143	0.0208	0.984	0.15
ln(*Pf*)	−10.8564	−0.3423	0.5893	−0.2302	0.1047	3.9064	1.8125	−0.6126	0.669	0.42
ln(*Pb*)	−11.3469	−0.1765	0.6888	−0.2263	0.0903	3.7022	1.7557	−0.5780	0.785	0.40

Designations here and further: ^(1)^ Dependent variables; ^(2)^ The constant corrected for logarithmic retransformation by Baskerville (1972); ^(3)^ adjR^2^—determination coefficient adjusted for the number of variables; ^(4)^ SE—standard error of the equations.

**Table 2 plants-09-01255-t002:** Contribution of independent variables of Equation (1) to the explanation of the variability of dependent variables, %.

ln(*Y*)	Independent Variables ^2^								
	ln*A*(I)	ln*V*(II)	ln*N*(III)	(ln*A*)·(ln*N)* (IV)	I + II + III + IV	ln(*Tm* + M) (V)	ln*PRm* (VI)	[ln(*Tm* + M)]·(ln*PRm*) (VII)	V + VI + VII
*Pinus*									
ln(*Pt*)	5.5	88.8	1.1	0.3	95.7	1.3	1.4	1.6	4.3
ln(*Pa*)	2.7	84.4	2.9	2.8	92.8	2.3	2.3	2.6	7.2
ln(*Pr*)	11.4	62.4	3.2	5.5	82.5	6.0	5.7	5.8	17.5
ln(*Ps*)	7.1	88.0	0.5	1.5	97.1	0.9	0.9	1.1	2.9
ln(*Pf*)	22.4	47.3	4.3	3.6	77.6	7.9	6.8	7.7	22.4
ln(*Pb*)	10.5	64.1	1.1	2.0	77.7	7.6	6.7	8.0	22.3
X ± σ ^1^	9.9 ± 6.9	72.5 ± 17.1	2.2 ± 1.5	2.6 ± 1.8	87.2 ± 9.0	4.3 ± 3.2	4.0 ± 2.7	4.5 ± 3.1	12.8 ± 9.0
*Picea*									
ln(*Pt*)	3.3	89.7	2.1	2.0	97.1	1.2	0.5	1.2	2.9
ln(*Pa*)	2.8	89.2	2.2	2.4	96.6	1.4	0.8	1.2	3.4
ln(*Pr*)	0.5	90.4	2.3	2.9	96.1	1.3	1.9	0.7	3.9
ln(*Ps*)	1.7	90.2	3.5	3.7	99.1	0.1	0.6	0.2	0.9
ln(*Pf*)	15.1	53.9	5.2	9.5	83.7	5.8	4.9	5.6	16.3
ln(*Pb*)	7.8	63.4	5.2	8.2	84.6	5.5	4.8	5.3	15.4
X ± σ	5.2 ± 5.4	79.5 ± 16.4	3.4 ± 1.5	4.8 ± 3.2	92.9 ± 6.8	2.6 ± 2.4	2.3 ± 2.1	2.4 ± 2.4	7.1 ± 6.8

^1^ X ± σ—mean ± standard deviation; ^2^ Designations of independent variables: see in characteristics of Equation (1).

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
