# Peer review of "Are There Differences in the Reaction of the Light-Tolerant Subgenus Pinus spp. Biomass to Climate Change as Compared to Light-Intolerant Genus Picea spp.?"

_plants, 2020, doi:10.3390/plants9101255_

Round 1

Reviewer 1 Report

Dear Authors

I like the paper, however please consider some remarks:

 L6- ...and" - who?

L18 – erase space before “So…"

L68 – erase dot after “…2017).” And continue with small letter “there”

L82_ delete space after “…expression”

L 101- add “is” after This

L101-102 – currently in European Scots pine forests there are only 2 sets of needles on trees (instead of 5-6 mentioned in the old literature), it may show morphological changes in forest under transition because of climatic changes.

L113- erase space after “...tundra.” and replace “is” by it.

L119-120 – please translate from Russian to English

L122 – change the order “…can change biomass”, erase “with” replacing it by “taking into account”

L136- delete space before C

Figure 1 has different font – unify

L182 – unify font of the link in blue

L183 – coma is not needed after “correction”

L233 – change to “it increases” or “it is/was increased”

L238-248 – probably better use pass tense “took”, “got”. This phragments looks like “material and methods” – place there or rephrase

L247 – use singular “temperature”

L298-303 – is not clear – better rephrase

L320 – at the end of this short sentence I suggests to add “otherwise we may also lose valuable genetic pools (Nowakowska et al., 2020).

Nowakowska, J. A., Hsiang, T., Patynek, P., StereÅ„czak, K., Olejarski, I., & Oszako, T. (2020). Health Assessment and Genetic Structure of Monumental Norway Spruce Trees during A Bark Beetle (Ips typographus L.) Outbreak in the BiaÅ‚owieża Forest District, Poland. Forests, 11(6), 647.  

L352-delete one dot

L359 - delete extra space between "to" and "predict"

L370 - add space after ":", and check it further in the references to keep the same layout

Author Response

Dear Editor, 

Please find the reply in the attached file.

Sincerely, 

Reviewer 2 Report

Dear Authors,

All my comments I included in the attached text.

Detailed comments

  • First, the title of the manuscript must be improved. The Authors use the terms: light-coniferous and dark-coniferous genus. I think that it would be much better to use light-tolerant and light-intolerant ones. Also, why you use Pinus And Picea spp. Shouldn’t be “spp.” for both?
  • Section “Materials and methods” should be also rewritten. Also, there should be information on the methods of biomass calculations.
  • Authors should consider to transfer tables (large ones) to supplementary materials. Tab. 1 and 2 – they represent results….I suppose.
  • Figures and their captions should be improved and unified where possible.
  • Section “Results and discussion” – should be rewritten and shorten. See attached text.

More remarks are included in pdf file attached.

Author Response

Dear Editor, 

Please find the replies in the attached file. 

Sincerely, 

Reviewer 3 Report

The article is focused on very interesting area of research on forest biomass versus global climate changes objective. Maybe the description of method which was used need to be little bit more deep . Moreover, I mean that in the chapter Methods the some introduction of results and polemics  with other authors is not common. Results are presented in the clearly way.
